# Nirmatrelvir/Ritonavir and Molnupiravir in the Treatment of Mild/Moderate COVID-19: Results of a Real-Life Study

**DOI:** 10.3390/vaccines10101731

**Published:** 2022-10-17

**Authors:** Ivan Gentile, Riccardo Scotto, Nicola Schiano Moriello, Biagio Pinchera, Riccardo Villari, Emilia Trucillo, Luigi Ametrano, Ludovica Fusco, Giuseppe Castaldo, Antonio Riccardo Buonomo

**Affiliations:** 1Department of Clinical Medicine and Surgery—Section of Infectious Diseases, University of Naples Federico II, 80131 Naples, Italy; 2Department of Molecular Medicine and Medical Biotechnologies, University of Naples Federico II, 80131 Naples, Italy; 3CEINGE—Advanced Biotechnologies, 80131 Naples, Italy

**Keywords:** COVID-19, SARS-CoV-2, molnupiravir, nirmatrelvir/ritonavir, hospitalization, adverse drug reactions

## Abstract

Molnupiravir and nirmatrelvir were the first available oral antivirals (OAs) active against SARS-CoV-2. Trials evaluating the efficacy of OAs involved patients unvaccinated and infected with variants different from those currently circulating. We conducted a retrospective study on patients with confirmed SARS-CoV-2 infection treated with OAs during the omicron surge in Italy in order to provide real-life data on the efficacy and safety of OAs during the omicron surge of the COVID-19 pandemic. Among 257 patients, 56.8% received molnupiravir, while 43.2% received nirmatrelvir/ritonavir. Patients in the molnupiravir group were older, had a lower body mass index, and had a higher rate of chronic heart disease than those treated with nirmatrelvir/ritonavir. Three hospitalizations were recorded in the molnupiravir (2.1%) group and one in the nirmatrelvir/ritonavir (0.9%) group. One patient treated with molnupiravir died. The median time to negativity was 8 days in the nirmatrelvir/ritonavir group vs. 10 days in the molnupiravir group, *p* < 0.01. We recorded 37 ADRs (mainly dysgeusia, diarrhea, and nausea) in 31 individuals (12.1%). Only two patients (0.8%) treated with molnupiravir terminated treatment due to ADRs. In conclusion, in a population of mostly vaccinated patients treated with OAs, we observed a low rate of hospitalization, death, and adverse drug reactions. These rates were lower than those reported in pivotal trials.

## 1. Introduction

COVID-19 is a disease caused by the betacoronavirus SARS-CoV-2 [1]. In late 2019 in Wuhan, China, the first instances of COVID-19 were discovered [2], and the virus quickly spread around the globe [3], causing more than 500 million infections and 6 million fatalities to date [4]. Antivirals against SARS-CoV-2 work by inhibiting viral replication and prevent, in most cases, the deterioration of patients toward a severe form of the disease [5]. Several antivirals active against COVID-19 are currently available. The first one to be widely used was remdesivir, a polymerase inhibitor that is administered intravenously. However, this drug has logistical issues because it requires intravenous injections [6]. Monoclonal antibodies are another category of drugs which inhibit virus entry into the host cell and are to be administered intravenously as well. However, the real revolution in the treatment of COVID-19 was the introduction of orally available antivirals, molnupiravir and nirmatrelvir, in clinical practice [7,8]. Molnupiravir acts by binding to the RNA-dependent RNA-polymerase, causing multiple errors leading to a “lethal mutagenesis” and finally blocking viral replication [9,10,11]. Nirmatrelvir acts by inhibiting the viral protease Mpro, which is essential for viral replication [12,13]. It is administered with ritonavir to boost pharmacokinetics and is a CYP3A4 inhibitor [12]. With respect to the registration trials of oral antivirals, it is noteworthy that patients infected with the omicron variant were not enrolled. However, molnupiravir and nirmatrelvir should not be affected by mutations that occur mainly in the spike protein because this protein is not the target of the two drugs. This theoretical consideration is supported by in vitro studies that show similar susceptibility to the two antivirals of BA.2.12.1, BA.4, and BA.5 variants compared to the original Wuhan strain [14]. This finding needs to be confirmed by in vivo data. Indeed, very few studies on the real-life efficacy of oral antivirals (OA) against the omicron variant of SARS-CoV-2 and in vaccinated individuals are currently available. The present study aims to provide real-life data on the occurrence of unfavorable outcomes among patients treated with molnupiravir and nirmatrelvir/ritonavir during the omicron surge of the COVID-19 pandemic.

## 2. Materials and Methods

This real-life retrospective study was conducted on all patients referred to the Unit of Infectious Diseases, University of Naples Federico II, Campania Region, Italy, between 18 February 2022 and 30 June 2022 with a diagnosis of SARS-CoV-2 infection who were treated with oral antivirals for COVID-19, namely, molnupiravir or nirmatrelvir/ritonavir.

Diagnosis of SARS-CoV-2 infection was made through an antigenic or PCR-based validated test. No exclusion criteria were set in order to provide real-life results not influenced by selection criteria. However, in Italy, the administration of OAs for COVID-19 is regulated by strict indications provided by the Italian Drug Agency (AIFA, Agenzia Italiana del Farmaco) [15]. In particular, patients with SARS-CoV-2 infection eligible for OA treatment are those not requiring hospitalization due to COVID-19 at the time of OAs first administration and at high risk for disease progression due to older age and comorbidities within 5 days from COVID-19 symptoms onset. Comorbidities conferring an increased risk of COVID-19 progression include active onco-hematological disease; chronic kidney disease (CKD); severe chronic lung disease; primary or acquired immunodeficiency; obesity (body mass index [BMI]) ≥ 30; severe cardiovascular disease; and uncontrolled diabetes mellitus.

All patients referred for OAs were screened for indication by a dedicated medical site staff, which was also responsible for the choice of treatment. All patients were screened for drug-to-drug interactions (DDI) between concomitant chronic treatments and nirmatrelvir/ritonavir. DDIs were screened using Medscape© drug interaction checker (available at: https://reference.medscape.com/drug-interactionchecker?faf=1&ecd=ppc_google_rlsa-traf_mscp_ref_md-6mo-lapsed_englang-general-int&gclid=Cj0KCQjwnP-ZBhDiARIsAH3FSRe95aSm3mQBjpvf5VmxKwcconML1ZcYB2rs3mhxmc_BKlATc2BsxO0aAn-AEALw_wcB, accessed on 15 September 2022). In the presence of significant DDIs, patients were treated with molnupiravir; otherwise, nirmatrelvir/ritonavir was the treatment of choice in most cases. OAs were administered for five days at a dosage of 300/100 mg of nirmatrelvir/ritonavir twice daily or 800 mg of molnupiravir twice daily. In case of nirmatrelvir/ritonavir administration, patients with an eGFR between 30 and 60 mL/min/m^2^ received 150/100 of nirmatrelvir/ritonavir twice daily. All treated patients with an eGFR lower than 30 mL/min/m^2^ received molnupiravir. All eligible patients were asked to sign a specific consent form, as also requested by AIFA, and to undergo a medical examination before receiving the assigned treatment (Day 0). All patients received detailed instructions on how to take OAs and were asked to autonomously assume treatment at home. Patients were also trained on identifying adverse drug reactions (ADR) possibly related to OAs and were asked to take note of all symptoms possibly occurring during and after treatment. They were also invited to contact the medical staff in case of necessity and to promptly notify them of the finding of a negative SARS-CoV-2 swab. Aside from spontaneous contact with included patients, the medical site staff performed a telephonic evaluation for all treated patients on Day 7 and Day 14 to investigate the occurrence of new COVID-19-related symptoms and possible ADRs. A COVID-19-related hospitalization was defined as the need for hospitalization in patients treated with OAs for SARS-CoV-2 infection, requiring oxygen supplementation treatment for progression of COVID-19-related symptoms. All causes of hospitalizations were recorded as well.

The study’s endpoints were to assess COVID-19-related and all-causes hospitalization rates and mortality rates among patients with SARS-CoV-2 infection treated with OAs according to AIFA criteria. The incidence of ADR was also recorded, as well as risk factors for hospitalization.

### Statistical Analysis

All the variables were tested for parametric/non-parametric distribution with the Kolmogorov–Smirnov test. Comparisons between categorical dichotomous variables were performed with the χ^2^ test (or with Fischer’s exact test when applicable), while comparisons between ordinary variables were conducted with the t-student test (parametric variables) or the Mann–Whitney’s U test (non-parametric variables). Hospitalization rate was reported as person-per-year (PPY) with 95% confidence interval (95CI). All the statistics were performed with IBM^©^ SPSS software, version 26.

## 3. Results

We enrolled 257 patients with SARS-CoV-2 infection who were treated with OAs for COVID-19 during the study period. Among these, 146 (56.8%) were treated with molnupiravir, while 111 (43.2%) received the nirmatrelvir/ritonavir combination. Treatment with OAs was started within 5 days from COVID-19-related symptoms’ onset; the median time from symptoms onset to treatment administration was 3 days (IQR: 2–4) in both treatment groups (*p* = 0.153). Clinical data of enrolled patients are shown in Table 1.

Patients treated with nirmatrelvir/ritonavir were younger compared with those treated with molnupiravir (*p* < 0.001) and showed a higher percentage of obesity among the comorbidities (26.1% vs. 15.7%, *p* < 0.05). On the contrary, patients treated with molnupiravir showed a higher percentage of chronic heart disease compared with those treated with nirmatrelvir/ritonavir (40.4% vs. 12.6%, *p* < 0.001). Patients with ≥2 comorbidities were more frequently treated with molnupiravir than with nirmatrelvir/ritonavir.

Throughout the 14-day follow-up, only 4 hospitalizations were recorded among the 257 patients treated with OAs (1.6%). The standardized rate of hospitalizations was 5.68 PPY. Most of the hospitalizations occurred among patients treated with molnupiravir (3, 2.1%; standardized rate: 7.5 PPY), while only one patient (0.9%) treated with nirmatrelvir/ritonavir was hospitalized (standardized rate: 3.28 PPY). All recorded hospitalization were related to COVID-19 symptoms; thus, there was no difference between COVID-19-related hospitalizations and all-causes hospitalizations.

Most patients who needed hospitalization were male (3/4, 75%) and were >75 years old; they all had a MASS score ≥ 2. Moreover, three patients (2.1%) were treated with molnupiravir, while one patient was treated with nirmatrelvir/ritonavir (0.9%) (*p* = 0.460). Similar rates of hospitalization were recorded in patients with ≥2 comorbidities (1.4%) compared with patients with none or one comorbidity (1.6%, *p* = 1.000). Only one patient treated with molnupiravir died.

The median time to obtain a negative SARS-CoV-2 swab in patients treated with OAs was of 8 days (IQR: 7–13), with a lower time in patients treated with nirmatrelvir/ritonavir (8 days; IQR: 6–11) compared with patients treated with molnupiravir (10 days; IQR: 7–17, *p* < 0.01).

With regard to the 10 unvaccinated patients, they mostly had 1 comorbidity (7, 70%), while 2 out of 10 (20%) had ≥2 comorbidities, and only 1 had no comorbidities. None of them required hospitalization due to COVID-19. The median time to obtain a negative SARS-CoV-2 swab among unvaccinated patients was 11 days (7–18), and it was similar to the time observed in patients who received SARS-CoV-2 vaccination (8 days; IQR: 7–13, *p* = 0.306.

With respect to ADRs, globally, we recorded 37 ADRs occurring in 31 of the 257 patients (12.1%), with 26 patients (10.1%) referring 1 ADR and 5 patients (1.9%) experiencing 2 ADRs. A total of 13 patients (8.9%) and 18 patients (16.2%) treated with molnupiravir and nirmatrelvir/ritonavir, respectively, reported at least one ADR (*p* = 0.075). The most common ADR was dysgeusia, reported by 14 patients (5.4%). Other recorded ADRs were: nausea (6 patients, 2.3%); diarrhea (6 patients, 2.3%); headache (4 patients, 1.6%); skin rash (2 patients, 0.8%); vomit (1 patient, 0.4%); dizziness (1 patient; 0.4%); and seizure (1 patient, 0.4%). Only two patients (0.8%), both treated with molnupiravir, discontinued treatment due to the occurrence of ADRs (seizures and dizziness, respectively). Dysgeusia was more commonly reported by patients treated with nirmatrelvir compared with those who received molnupiravir (9.0% vs. 2.7%, *p* < 0.05; Table 2).

## 4. Discussion

In this prospective, real-life cohort study, 257 patients with SARS-CoV-2 infection were treated with oral molnupiravir or oral nirmatrelvir/ritonavir according to the indications provided by AIFA [15]. The introduction of OAs was a revolution in the treatment of COVID-19. However, the pivotal trials which led to their approval enrolled patients that are different from those who could benefit from these drugs, mainly with respect to vaccination status and viral variant.

What do we know from clinical trials? Molnupiravir efficacy and safety were assessed in the MOVe-OUT, a phase 2/3 randomized, double-blind, placebo-controlled trial. This trial enrolled patients with a SARS-CoV-2 infection with symptoms onset within 5 days and at least one risk factor for the development of severe illness from COVID-19: age >60 years; active cancer; chronic kidney disease; chronic obstructive pulmonary disease; obesity; and serious heart conditions. The most prevalent variant was delta, followed by mu and gamma. Notably, no omicron cases were studied (the trial ended before the first case of the omicron variant) [16]. All participants in the trial were not vaccinated. Molnupiravir met the superiority criterion vs. placebo: the patients receiving molnupiravir had a lower risk of death or hospitalization at Day 29 after administration compared to those receiving placebo (6.8% vs. 9.7%; difference, −3.0 percentage points; 95% CI, −5.9 to −0.1) [16]. Another trial has been conducted to assess the utility of molnupiravir in patients requiring in-hospital treatment for COVID-19 with symptom onset ten or fewer days before randomization. In this context, the drug failed to manifest any clinical benefit [17].

The efficacy of the combination of nirmatrelvir and ritonavir was evaluated in the EPIC-HR trial: a double-blind, randomized, placebo-controlled trial [18]. This trial enrolled adults older than 18 years, with at least one risk factor for progression to severe disease, a confirmed SARS-CoV-2 infection and symptom onset not earlier than 5 days. All patients in the trial were not vaccinated and did not experience an earlier SARS-CoV-2 infection. The authors of the trial did not report details on SARS-CoV-2 variants. However, the trial enrolment finished just a few days after the first omicron cases were diagnosed in the world [19], so omicron prevalence among the trial subjects should be minimal, if any. The primary objective of the trial was to assess the drug’s efficacy in preventing COVID-19-related hospitalization or death. The drug successfully reached the endpoint, and was superior to the placebo (difference of −5.81 percentage points in 95% CI, −7.78 to −3.84; *p* < 0.001; relative risk reduction, 88.9%) [18].

In our study, given the study period (February–June 2022), included patients were assumed to mostly harbor the omicron variant of SARS-CoV-2. In consideration of the eligibility criteria for OA treatment, most patients included in this analysis were older or had significant comorbidities. This evidence represents a substantial difference with clinical trials investigating the efficacy and safety of OAs for SARS-CoV-2. In fact, the median age of enrolled patients in the pivotal trial of nirmatrelvir/ritonavir and molnupiravir was 45 (95CI: 18–86) and 42 (95CI: 18–90) years, respectively [18]. Furthermore, in both these clinical trials, an increased Body Mass Index was the most common risk factor for COVID-19 progression among the enrolled patients. Obesity was indeed the most frequent comorbidity of patients treated with molnupiravir in the MOVe-OUT trial, while 80.5% of patients enrolled in the EPIC-HR study showed a BMI of 25 or above. On the contrary, a significant percentage of patients included in our real-life cohort had primary or iatrogenic immunodeficiency mostly related to active hematological disease and immunosuppressant treatment (49.3% in the molnupiravir group, 59.5% in the nirmatrelvir/ritonavir group). Only a paucity of patients in the EPIC-HR and in the MOVe-OUT trials had immunodeficiency, overall making the population of the two trials considerably different from the real-life population of our study. Results from other real-life cohorts and retrospective studies assessing the safety and efficacy of OAs for SARS-CoV-2 also refer to populations with a very low rate of patients with immunodeficiency, with obesity generally representing the most common risk factor for COVID-19 progression [20,21]. These differences must be taken into account when efficacy and safety data of OAs from this cohort are reported.

Despite the high rate of patients with advanced age and severe comorbidities, the outcome was favorable in most cases. Only four patients (1.6%, 5.68 PPY) indeed needed hospitalization, and only one patient (0.4%) died, without significant differences between the two antivirals. Even without an untreated control group, it is noteworthy that the rates of hospitalization and death were considerably lower compared with those reported in clinical studies and by governmental agencies. Results from the COVID-NET network indeed showed that, in a period of omicron predominance, the proportion of hospitalized, vaccinated adults with COVID-19 peaked at 13.4% among the general population [22]. We underline that our cohort includes a high number of patients with immunodeficiency who are, among at-risk patients, those particularly at risk for severe COVID-19, hospitalization, and death [23,24,25,26].

Additionally, comparing our data with those that emerged from pivotal trials, it is noteworthy that the rate of hospitalization reported among patients treated with molnupiravir in our cohort was consistently lower compared with the hospitalization rate reported in the MOVe-OUT trial (2.1% vs. 7.3%) [16], while the rate of hospitalized patients treated with nirmatrelvir/ritonavir was similar to the one recorded in the EPIC-HR study (0.9% vs. 0.72%) [18]. However, these results should be interpreted in light of the characteristics of the population of our study, which is made up of nearly all vaccinated, tough frail subjects. In fact, SARS-CoV-2 vaccination likely influenced the overall outcome observed in the study population. However, due to the paucity of data on OAs’ efficacy in vaccinated patients, both vaccinated and unvaccinated patients can currently be treated with OAs. The real advantage of the treatment in this population remains largely unclear.

Very few real-life studies have been available so far. A large study has been conducted in Israel on a healthcare provider’s database of patients diagnosed with SARS-CoV-2 infection between January and February 2022 who were at high risk for severe COVID-19 and had no contraindications for nirmatrelvir/ritonavir use [20]. Of the total sample of over 180,000 eligible patients, 4732 (2.6%) were treated with nirmatrelvir/ritonavir. The authors found a significant decrease in the rate of severe COVID-19 (adjusted HR 0.54; 95% CI, 0.39–0.75) or death (adjusted HR 0.20; 95% CI, 0.17–0.22) for the patients treated with nirmatrelvir/ritonavir with respect to untreated patients [20]. The majority of patients (75.1%) were vaccinated against SARS-CoV-2. Information about the variants was not available, but according to the timeframe in which the study was performed, it is likely that at least in the second half of the study, BA.2 was the predominant variant [20]. Another retrospective study was conducted in Hong Kong. This study evaluated all-cause mortality in hospitalized patients with a mild form of COVID-19 receiving molnupiravir or nirmatrelvir/ritonavir. Each group was matched in a ratio of 1:1 with untreated controls. The study found that the use of both OAs was associated with a reduction in all-cause mortality: molnupiravir HR = 0.48 (HR = 0.48, 95% CI = 0.40–0.59, *p* < 0.0001); nirmatrelvir/ritonavir: HR = 0.34 (95% CI = 0.23–0.50, *p* < 0.0001) [27]. Mortality rates were 8.1% and 15.9% in the molnupiravir and control group, respectively, and 3.6% and 10.3% in the nirmatrelvir/ritonavir and control group, respectively. However, different from our cohort, only a minority of the patients in that study were vaccinated: 6.2% of those receiving molnupiravir and 10.5% of those receiving nirmatrelvir/ritonavir.

Another interesting result of our study was the tolerability of oral antivirals. Despite the old age of included patients and the presence of severe comorbidities, the rate of recorded ADRs was low. In fact, only 12.1% of patients treated with OAs reported an ADR (8.9% in the molnupiravir group and 16.2% in the nirmatrelvir/ritonavir group). Furthermore, the rates of patients with ADR were lower compared with data from the MOVe-OUT trial (30.4% of patients with at least one ADR in the molnupiravir group) [16] and of the EPIC-HR trial (22.6% of patients with at least one ADR in the nirmatrelvir/ritonavir group) [18]. Interestingly, the most common ADR reported from patients in this cohort was dysgeusia. This ADR was more frequent among patients treated with nirmatrelvir/ritonavir compared with those treated with molnupiravir (9.0% vs. 2.7%, *p* < 0.05), while the occurrence of dysgeusia was not reported in the MOVe-OUT trial [16]. Otherwise, in the EPIC-HR trial, 5.3% of patients treated with nirmatrelvir/ritonavir experienced dysgeusia [18], confirming a possible association between dysgeusia and treatment with nirmatrelvir/ritonavir. We acknowledge that clinical trials usually assess ADRs differently and often more thoroughly compared to real-life studies. However, we underline the very low rate of discontinuation of our cohort, which reflects the excellent tolerability of both antivirals.

The strength of this work is the availability of real-life data in vaccinated patients during the omicron surge and the assessment of safety in a cohort of older patients affected by severe comorbidities, in particular immunodeficiency. On the contrary, a consistent limit of this study is represented by the absence of a control group of untreated patients. However, it must be said that denying OA treatment to frail patients with SARS-CoV-2 in a real-life scenario should be considered unethical. Another limitation of our study is represented by the absence of a systematic follow-up at the medical site to assess viral clearance. Data on follow-up swabs for SARS-CoV-2 were mostly spontaneously reported by the included patients.

## 5. Conclusions

In conclusion, results from this real-life retrospective study showed a very low rate of hospitalization, death, and ADRs among mostly vaccinated patients with significant comorbidities who were treated with oral molnupiravir or nirmatrelvir/ritonavir for SARS-CoV-2 infection. Rates of such outcomes were lower compared with those reported in randomized controlled trials and in other real-life experiences with OAs for SARS-CoV-2 infection.

## Figures and Tables

**Table 1 vaccines-10-01731-t001:** Clinical characteristics of patients with SARS-CoV-2 infection who received oral antiviral therapy.

	Total (n = 257)	Molnupiravir (n = 146)	Nirmatrelvir/Ritonavir (n = 111)	*p*-Value
**Sex (n, %)**				
- Male	124 (48.2)	77 (52.7)	47 (42.3)	0.098
- Female	133 (51.7)	69 (47.2)	64 (57.6)	
**Age (years; median, IQR)**	64 (52–75)	70 (59–79)	60 (40–67)	<0.001
**Age (years; n, %)**				
- 19–65	140 (54.5)	62 (42.5)	78 (70.3)	<0.001
- >65	117 (45.5)	84 (57.5)	33 (29.7)	
**Comorbidities**				
- BMI ≥ 30	52 (20.2)	23 (15.7)	29 (26.1)	<0.05
- Chronic kidney disease	14 (5.4)	11 (7.5)	3 (2.7)	0.091
- Uncontrolled diabetes mellitus	18 (7.0)	13 (8.9)	5 (4.5)	0.171
- Immunodeficiency	139 (54.1)	72 (49.3)	66 (59.5)	0.085
- Chronic heart disease	73 (28.4)	59 (40.4)	14 (12.6)	<0.001
- Chronic liver disease	4 (1.6)	4 (2.7)	0 (0.0)	0.136
- Severe chronic lung disease	24 (9.3)	18 (12.3)	6 (5.4)	0.059
- Neurodegenerative disorder	9 (3.5)	7 (4.8)	2 (1.8)	0.472
**Number of comorbidities**				
- 0	10 (3.9)	4 (2.7)	6 (5.4)	0.336
- 1	175 (68.1)	88 (60.3)	87 (78.4)	<0.01
- 2	61 (23.7)	45 (30.8)	16 (26.2)	<0.01
- ≥3	11 (4.3)	9 (6.2)	2 (1.8)	0.121
**MASS score ^#^ (median, IQR)**	3 (2–3)	3 (2–3)	3 (2.3)	0.694
**SARS-CoV-2 vaccination received * (n, %)**	247 (96.1)	138 (94.5)	109 (98.2)	0.195

^#^ Monoclonal Antibody Screening Score (MASS) assigned a score to each of the original US FDA EUA criteria (released November 2020), as follows: age ≥ 65 years (2), BMI ≥ 35 Kg/m^2^ (2), diabetes mellitus (2), chronic kidney disease (3), cardiovascular disease in a patient ≥ 55 years (2), chronic respiratory disease in a patient ≥ 55 years (3), hypertension in a patient ≥ 55 years (1), and immunocompromised status (3). Maximum score is 18. * at least 2 doses received.

**Table 2 vaccines-10-01731-t002:** Side effects recorded among the included patients and according to treatment received.

	Molnupiravir (n = 146)	Nirmatrelvir/Ritonavir (n = 111)	*p*-Value
**Diarrhea**	4 (2.7%)	2 (1.8%)	0.701
**Dizziness**	1 (0.7%)	0 (0.0%)	1.000
**Dysgeusia**	4 (2.7%)	10 (9.0%)	<0.05
**Headache**	2 (1.7%)	2 (1.8%)	1.000
**Nausea**	2 (1.7%)	4 (3.6%)	0.407
**Seizure**	1 (0.7%)	0 (0.0%)	1.000
**Skin rash**	1 (0.7%)	1 (0.9%)	1.000
**Vomit**	1 (0.7%)	0 (0.0%)	1.000
**Discontinuation of treatment**	2 (1.7%)	0 (0.0%)	0.507

## Data Availability

The data presented in this study are available on request from the corresponding author.

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
