# Peer review of "Nirmatrelvir/Ritonavir and Molnupiravir in the Treatment of Mild/Moderate COVID-19: Results of a Real-Life Study"

_vaccines, 2022, doi:10.3390/vaccines10101731_

Round 1
Reviewer 1 Report
In the manuscript by Gentile et al (vaccines-1947298), the authors conducted a retrospective study on COVID-19 patients under oral antivirals treatment and found that vaccinated patients are benefit from oral antivirals treatment with lower rate of hospitalization and death. This study provides important information on the efficacy of oral antivirals for COVID-19 patients during Omicron emerging.
Specific comments:
1. As we know that molnupiravir/nirmatrelvir should be given within 5 days of symptom onset. The authors should include the date information of the patient’s symptom onset and the date which oral antivirals were given to patients. Is there difference between monupiravir and Nirmatrelvir?
2. In the manuscript, “COVID-19”, “Covid 19” or “Covid-19” were used. Please keep it consistent throughout the manuscript by using “COVID-19”.
Author Response
Dear Editors,
Thank you very much for considering our paper for publication in Vaccines. Please find attached the point-by-point replies to the reviewer as well as a "tracked" version of our revised manuscript.
Kind Regards
Riccardo Scotto M.D., Ph. D.
Department of Clinical Medicine and Surgery
Section of Infectious Diseases
University of Naples Federico II
Reviewer 2 Report
The study attempts to describe a real-life scenario on the use of oral antiviral (Nirmatrelvir and Molnupiravir) in the treatment of mild/moderate COVID-19. Given the increasing advocacy for use of oral antivirals, studies of this nature are important to strengthen the limited data available on the safety and efficacy of oral antivirals.
Although important, the authors will need to address some major issues in the manuscript.
Major comments
The general aim of the manuscript was to provide data on the efficacy and safety of two antivirals during the omicron surge in Italy. The study design employed however seems to only measure the prevalence or frequency of hospitalisation, adverse drug reactions and death among patients receiving this medication. Authors did not use any comparator such as controls in their design or extraction of the retrospective data. I believe some patients were given standard treatment for COVID-19 while others opted to take the oral antivirals. If so, why didn't authors include this in their analysis? The authors determination of efficacy and safety of the OAs, without the use of controls, makes their findings a bit vague and difficult to appreciate. The findings do not address the objectives mentioned in line (57-58). It will be good for authors to include additional information on this instead of comparing prevalence rates in their cohort to other published studies within the context of clinical trials. Its difficult to appreciate the contribution of this work to knowledge if the right data and design is not employed in this report.
2. Authors recruited both vaccinated and unvaccinated subjects in their study( 247 vrs 10) but did not present information on the status of the unvaccinated persons. Authors' conclusion of low rates of hospitalisations and deaths could clearly be due to the taking of two full doses of vaccines which are known to have similar effects. How do authors address these confounder? Is it the vaccine which is causing the low hospitalisations or the oral antivirals? It will be worthwhile if authors have more data to present on unvaccinated individuals so we can know the true effect of the drug.
3. Authors again indicated that OAs for COVID-19 is only given to patients not requiring hospitalisation (line 67=71) as per Italian Drug Agency policy. It is therefore a bit surprising that authors will mention hospitalisation as one of their primary endpoints when those requiring for hospitalisation were excluded in the enrollment of patients. Could it be that this is what accounted for the low hospitalisation and not the OAs?
4. Line 79: Authors mentioned patients were screened for drug-drug interactions but did not explain the kind of screening done? This should have been indicated in their methods. I'm not sure whether measuring eGFR qualified as a test for drug-drug interactions. The authors should explain this.
Author Response
Point 1: The general aim of the manuscript was to provide data on the efficacy and safety of two antivirals during the omicron surge in Italy. The study design employed however seems to only measure the prevalence or frequency of hospitalisation, adverse drug reactions and death among patients receiving this medication. Authors did not use any comparator such as controls in their design or extraction of the retrospective data. I believe some patients were given standard treatment for COVID-19 while others opted to take the oral antivirals. If so, why didn't authors include this in their analysis? The authors determination of efficacy and safety of the OAs, without the use of controls, makes their findings a bit vague and difficult to appreciate. The findings do not address the objectives mentioned in line (57-58). It will be good for authors to include additional information on this instead of comparing prevalence rates in their cohort to other published studies within the context of clinical trials. Its difficult to appreciate the contribution of this work to knowledge if the right data and design is not employed in this report.
Response 1: We totally agree with the reviwer and we are well aware of what the referee claims. As stated in the discussion section, we ackowledge that the lack of a control group represents a consistent limitation of our study. However, it must be said that it is difficult to choose a reliable comparator in a real-life study on oral antivirals for SARS-CoV-2 infection, at present. All the patients referring to our unit of infectious disease who were eligible to OAs (according to Italian Drug Agency) were indeed treated, as OAs must be considered the standard treatment in this population (elderly and fragile patients with comordbidities). On the contrary, the very few of them who refused to receive OAs could not be included in the analysis due to missing informed consent. Having said that, the only option for a control group would be a standard symtomatic treatment among patients who were not eligible to OAs according to Italian Drug Agency (patients aged < 60 years with no comorbidities). It should appear clear to the reviewer that the latter option would have led to significant biases, given the profound differences in clinical charachteristics between eligible and non-eligible patients to OAs.
Point 2: Authors recruited both vaccinated and unvaccinated subjects in their study( 247 vrs 10) but did not present information on the status of the unvaccinated persons. Authors' conclusion of low rates of hospitalisations and deaths could clearly be due to the taking of two full doses of vaccines which are known to have similar effects. How do authors address these confounder? Is it the vaccine which is causing the low hospitalisations or the oral antivirals? It will be worthwhile if authors have more data to present on unvaccinated individuals so we can know the true effect of the drug
Response 2: We thank the reviwer for the suggestion. However, we strongly believe that a real-life study should report data on an actual clinical scenario. In this specific scenario, there are no limits to OAs elegibility according to vaccination status. In the everyday clinical practics after 2 years of pandemic, and after more than a year from vaccines availability, both vaccinated and non-vaccinated patients with SARS-CoV-2 infection refer to specialized clinicians to receive early treatment with OAs, and most of them are luckily vaccinated. Did the SARS-CoV-2 vaccination have an impact on the outcome? Yes, of course it did! But these are indeed the characteritics of patients who are treated with OAs nowadays, so clinicians should be interested in the efficacy of OAs among patients who are MOSTLY vaccinated. Randomized clinical studies on OAs for SARS-CoV-2 infection have been already conducted, but we need more real-life data, without the confounders of an “assembled” population. We stated that in our methods section: “No exclusion criteria were set, in order to provide real-life results not influenced by selection criteria”. It must also be said that patients eligible to OAs must be often considered immunocompromised subjects in whom the efficacy of vaccination can be impaired. For instance, in our cohort, more than 50% of patients had immunodeficiency (mostly related to onco-haematoligical diseases), and they were thus at high risk for severe COVID-19 and hospitalization, regardless of the vaccination status.
Having said that, as the number of unvaccinated patients is very small (10%, 3.9%) we just added this sentence in result section: “With regards to the 10 unvaccinated patients, they mostly had 1 comorbidity (7, 70%), while 2 out of 10 (20%) had ≥ 2 comorbidities and only one had no comorbidities. None of them required hospitalization due to COVID-19. The median time to obtain a negative SARS-CoV-2 swab among unvaccinated patients was 11 days (7-18) and it was similar to the time observed in patients who received SARS-CoV-2 vaccination (8 days; IQR: 7-13, p=0.306)”
Point 3. Authors again indicated that OAs for COVID-19 is only given to patients not requiring hospitalisation (line 67=71) as per Italian Drug Agency policy. It is therefore a bit surprising that authors will mention hospitalisation as one of their primary endpoints when those requiring for hospitalisation were excluded in the enrollment of patients. Could it be that this is what accounted for the low hospitalisation and not the OAs?
Response 3: In all studies carried out with the use of antivirals in mild/moderate form, hospitalization due to COVID-19 was one of the endpoints for the evaluation of efficacy. It is indeed widely known that the aim of early administration of antivirals is to reduce the risk of hospitalization and progression towards severe forms of COVID-19. OAs must be administered in patients that not require hospitalization, but these patients may subsequently require hospitalization despite the treatment, of course.
To better clarify, we revised methods section as follows: “patients with SARS-CoV-2 infection eligible for OA treatment are those not requiring hospitalization due to COVID-19 at the time of OAs first administration”
Point 4: Line 79: Authors mentioned patients were screened for drug-drug interactions but did not explain the kind of screening done? This should have been indicated in their methods. I'm not sure whether measuring eGFR qualified as a test for drug-drug interactions. The authors should explain this.
Response 4: In fact, eGFR has nothing to do with drug-drug interactions., Actually, most clinicians check for interaction using specific apps for smartphones.. The app receives a list of drugs and gives the clinician an alert in case of interactions. We revised the following sentence in method section to clarify this point: “All patients were screened for drug-to-drug interactions (DDI) between concomitant chronic treatments and nirmatrelvir/ritonavir, using specific website or app-based tools.”.
Round 2
Reviewer 2 Report
The authors have responded to some of the concerns raised. However, there are other issues that should still be addressed:
1. Response to query 1: Authors have mentioned that their study analysed real-life scenarios that occurred in their clinical setting. Having read their response, it will be appropriate for the authors to amend the objectives indicated in their introduction as assessing "efficacy and safety". This is basically a descriptive study analysing the frequency of events among a cohort of patients who took OAs. The use of "efficacy and safety" will require a hypothesis-driven analysis which is not provided in this study.
2. Response to query 2: If authors acknowledge that vaccination could have resulted in the outcome provided, then this needs to be stated in the discussion. The conclusion suggested OA's were solely responses for the low hospitalisations recorded without acknowledging the impact of the vaccines.
Response to query 4: Authors should give reference to website and tools used for assessing the drug-drug interactions.
Author Response
Response to query 1: Authors have mentioned that their study analysed real-life scenarios that occurred in their clinical setting. Having read their response, it will be appropriate for the authors to amend the objectives indicated in their introduction as assessing "efficacy and safety". This is basically a descriptive study analysing the frequency of events among a cohort of patients who took OAs. The use of "efficacy and safety" will require a hypothesis-driven analysis which is not provided in this study.
Response 1: We would like to thank the reviewer for this comment. We revised the mentioned sentence of the introduction section, that now is as follows: “The present study aims to provide real-life data on the occurrence of unfavorable outcomes among patients treated with Molnupiravir and Nirmatrelvir/ritonavir during the omicron surge of COVID-19 pandemic.”
Response to query 2: If authors acknowledge that vaccination could have resulted in the outcome provided, then this needs to be stated in the discussion. The conclusion suggested OA's were solely responses for the low hospitalisations recorded without acknowledging the impact of the vaccines.
Response 2: We took the point. We included the following sentence in the discussion section: “In fact, SARS-CoV-2 vaccination likely influenced the overall outcome observed in the study population. However, due to the paucity of data of OAs efficacy in vaccinated patients, both vaccinated and unvaccinated patients can be currently treated with OAs. The real advantage of the treatment in this population remains largely unclear”.
We also revised our conclusion as follows: “In conclusion, results from this real-life retrospective study showed a very low rate of hospitalization, death and ADRs among mostly vaccinated patients with significant comorbidities who were treated with oral molnupiravir or nirmatrelvir/ritonavir for SARS-CoV-2 infection.
Response to query 4: Authors should give reference to website and tools used for assessing the drug-drug interactions.
Response 4: We revised the following sentence in the methods section: “All patients were screened for drug-to-drug interactions (DDI) between concomitant chronic treatments and nirmatrelvir/ritonavir. DDIs were screened using Medscape© drug interaction checker (available at: https://reference.medscape.com/drug-interactionchecker?faf=1&ecd=ppc_google_rlsa-traf_mscp_ref_md-6mo-lapsed_englang-general-int&gclid=Cj0KCQjwnP-ZBhDiARIsAH3FSRe95aSm3mQBjpvf5VmxKwcconML1ZcYB2rs3mhxmc_BKlATc2BsxO0aAn-AEALw_wcB)”